# Foreseeing Privacy Threats from Gradient Inversion Through the Lens of Angular Lipschitz Smoothness

## Abstract

Recent works proposed server-side *input recovery attacks* in federated learning (FL), in which an honest-but-curious server can recover clients' data (e.g., images) using shared model gradients, thus raising doubts regarding the safety of FL. However, the attack methods are typically demonstrated on only a few models or focus heavily on the reconstruction of a single image, which is easier than that of a batch (multiple images). Thus, in this study, we systematically re-evaluated state-of-the-art (SOTA) attack methods on a variety of models in the context of batch reconstruction. For a broad spectrum of models, we considered two types of model variations: *implicit* (i.e., *without any change in architecture*) and *explicit* (i.e., *with architectural changes*). Motivated by the re-evaluation results that the quality of reconstructed image batch differs per model, we propose *angular Lipschitz constant of a model gradient function with respect to an input* as a measure that explains the vulnerability of a model against input recovery attacks. The prototype of the proposed measure is derived from our theorem on the convergence of attackers' gradient matching optimization, and re-designed into the *scale-invariant* form to prevent trivial server-side loss scaling trick. We demonstrated the predictability of the proposed measure on the vulnerability under recovery attacks by empirically showing its strong monotonic correlation with not only loss drop during gradient-matching optimization but also the quality of the reconstructed image batch. We expect our measure to be a key factor for developing client-side defensive strategies against privacy threats in our proposed realistic FL setting called *black-box* setting, where the server deliberately conceals global model information from clients excluding model gradients.

## 1   Introduction

Federated learning (FL) is a cooperative machine learning between clients as local trainers and a central server as a global aggregator [14, 21]. Participants in FL cannot access raw data from others and only communicate with one another through gradients, which were believed to leak little information of the original data in the past.

However, recent studies [31, 30, 6, 26, 12] challenge *inverting* gradients back to original data, suggesting that there is potential for an honest-but-curious server to attack by sneakily recovering clients' data from gradients in FL. Their algorithms, so-called gradient inversion attacks, aim at optimizing input variables (e.g., images) to match the given gradients under the condition of fixed model weights. For better reconstruction quality, state-of-the-art (SOTA) attacks assume that both batch normalization (BN) [11] layers' statistics and private labels are known [6, 26, 12, 8]. However,

they are demonstrated on a limited range of global models. Thus, we systematically re-evaluated SOTA gradient inversion attacks on a variety of models in the context of *batch (or multiple images) reconstruction*, the recovery of input batch from the averaged gradients over itself, which is more difficult to solve than single image reconstruction, the recovery of single image from its gradient. In this paper, two kinds of model variations are considered, namely *implicit* and *explicit*.

Implicit model variations refer to a collection of different models with the same architecture. In this paper, we consider two types of implicit model variations: *BN modes* and *training epochs*.

- As mentioned previously, SOTA gradient inversion attack methods are demonstrated on models with BN layers to assume shared BN statistics. Note that there are two modes of a BN layer, namely, *train mode* and *eval mode*. In the reality of FL, the server can choose any mode among them. Therefore, we re-evaluated SOTA attacks by considering both modes of BN. This paper is the first to consider BN modes for the evaluation of gradient inversion attacks. We empirically found that the quality of reconstructed batch significantly changes by switching BN modes even for the same model weights.

- By reflecting the reality that clients can encounter global model from the server at any time, we consider models with different training epochs for the re-evaluation. This scheme extends the scope of previous works' training epoch choices of *black-and-white* manner: zero training epoch (untrained) and maximum training epochs (fully trained). We empirically found that the best reconstruction result was usually found at earlier training epochs, not untrained nor fully trained, thus raising the need to expand the evaluation criterion for attack methods.

Meanwhile, explicit model variations are more straightforward than implicit model variations as they only involve architectural changes. In this study, we consider two types of explicit model variations: *skip connections* and *channel size*.

- Residual networks (ResNets) [9] are frequently employed in previous works [26, 6, 31, 12] even for batch reconstruction, while networks *without skip connection* are introduced for only for the recovery of single image from its gradient [6]. Therefore, we explored how a skip connection affects the quality of SOTA gradient inversion attacks in the context of batch reconstruction. Our empirical findings suggest that models without skip connection are more robust against the gradient inversion attack than residual networks.

- The reconstruction quality is known to increase with the number of channels, but this property is demonstrated on single image reconstruction [30, 6]. Thus, we recap how the number of channels affects the attack quality in the context of batch reconstruction.

By re-evaluating SOTA attacks in a variety of models, we found that the vulnerability against gradient inversion attack significantly differs per model, implying the need of more strict evaluation criteria for attack methods. Then, clients are required to judge whether a shared model from the server is safe or not *before sending* locally computed gradients back for their privacy. In this study, we consider two settings on the transparency of global model information to clients: *white-box* and *black-box*. In a white-box setting, clients have an absolute control over global model such as the server; thus, clients can directly apply SOTA attacks to the model to assess its vulnerability.

On the other hand, a *black-box* setting only allows clients control over model gradients to restrict access to the global model possibly due to companies' secrets. For the client-side measurement of privacy leakage in this practical and difficult setting, we propose *angular Lipschitz constant of model gradients with respect to an input* as a predictive measure for the quality of reconstructed samples inverted from model gradients.

This measure is derived from our theorem in Sec. 4 that an attacker's gradient matching loss function drops more abruptly with a smaller $L$ in a particular range, where $L$ is Lipschitz constant of model gradients with respect to an input. However, using $L$ as a measure for privacy leakage would be inappropriate as $L$ can be any nonnegative value by loss function scaling. Therefore, inspired by scale-invariant cosine similarity loss function, we propose the angular Lipschitz constant, a *loss scaling-invariant* alternative to $L$. We experimentally found that both measure motonically correlates

with not only total loss drop during an attacker's optimization but also the reconstruction quality than the norm of gradients. These findings are expected to support the construction of client-side defense algorithms particularly for *black-box* setting, where only model gradients are given to clients as minimal information of the model as described in Fig. 5.

## 2 Prior Art in the Gradient Inversion Attack

Given the neural network function $f_w : \mathbb{R}^{b \times d} \to \mathbb{R}^{b \times c}$ ($w$, $b$, $d$, $c$ being the model weights, batch size, image size, and the number of classes, respectively), and the gradient $g^* = \frac{\partial \mathcal{L}(f_w(x^*), y^*)}{\partial w}$ computed with ground truth input batch $(x^*, y^*) \in \mathbb{R}^{b \times d} \times \mathbb{R}^b$ ($x^*$, $y^*$ being the image batch, and corresponding label batch) and the loss function $\mathcal{L} : \mathbb{R}^{b \times c} \times \mathbb{R}^b \to \mathbb{R}$ (e.g., cross-entropy loss), the goal of gradient inversion attack is to reconstruct an image batch $x \in \mathbb{R}^{b \times d}$, a resemblance of ground truth image batch $x^*$. In the context of federated learning (FL), $f_w$ is the global model, and $g^*$ is the gradient computed from a client. Then, a honest-but-curious server aims to recover the client's private data $x^*$.

A general method to tackle the problem of inverting gradients is to solve an optimization problem formulated as follows:

$$\arg\min_{x,y} \mathcal{L}_{grad}\big(\frac{\partial \mathcal{L}(f_w(x), y)}{\partial w}, \frac{\partial \mathcal{L}(f_w(x^*), y^*)}{\partial w}\big) + \alpha_{prior} \mathcal{R}_{prior}(x), \tag{1}$$

where $\mathcal{L}_{grad} : \mathbb{R}^N \times \mathbb{R}^N \to \mathbb{R}$ (N is the size of weights $w$) is the loss function for gradient matching (which closes the distance between current gradients and target gradients), $\mathcal{R}_{prior} : \mathbb{R}^{b \times d} \to \mathbb{R}$ is the regularization loss for image prior, with $\alpha_{prior}$ being its coefficient.

Prior to the advent of packages for automatic differentiation, the gradient term $g = \frac{\partial \mathcal{L}(f_w(x), y)}{\partial w}$ was computed as a function of $(x, y)$ in a closed form. For the computation to be tractable, $\mathcal{L}_{grad}$ was set to a squared loss ($\mathcal{L}(g, g^*) = ||g - g^*||_2^2$), and $f_w$ was also slightly modified from the original design of contemporary neural networks. For example, ReLU activation functions were replaced with Sigmoid, and all the strides in convolution modules were excluded from the original ResNet in [31]. Consequently, the choice of $f_w$ was limited.

Currently, with the advantages of automatic differentiation [22] and advanced deep learning optimization algorithms [13, 23, 5], solving for optimization problem in (1) becomes tractable for most contemporary deep neural networks without the need for modification. Further, the gradient matching loss is selected in a broad range from cosine similarity loss ($\mathcal{L}(g, g^*) = 1 - \frac{<g, g^*>}{||g|| ||g^*||}$) [6, 12, 10, 26] to L2 loss ($\mathcal{L}(g, g^*) = ||g - g^*||_2^2$) [31, 29, 26]. The liberation from the limited choice of loss functions and neural network architectures became the trigger of state-of-the-art attack methods.

State-of-the-art attack methods provide several assumptions which enable the baseline, which is only gradient-based, to be expanded.

First, the server is supposed to know the private labels of clients' images. Currently, estimating $x$ and $y$ becomes a sequential process, in which $y$ is estimated first, after which $x$ is approximated with the estimated $y = y^*_{approx}$ given. Rather than jointly learning $x$ and $y$ in (1), prior works suggest estimating $y$ directly by seeing the gradients from ground truth data $g^*$ before optimization [26, 29]. Therefore, the problem of estimating labels from gradients is separated from the original optimization problem in (1) [3, 25, 16] and some works, which focus on reconstruction of images rather than labels, assume that private labels are known [6, 12].

Second, the local batch statistics $\{\mu_l(x^*; w), \sigma_l^2(x^*; w)\}_{l=1}^M$ ($\mu_l(x^*; w)$, $\sigma_l(x^*; w)$, and $M$ being the batch mean of the $l^{th}$ batch normalization layer, batch standard deviation of the $l^{th}$ batch normalization (BN) layer, and number of the BN layers, respectively), computed with client's data batch, is given to the server. This assumption reflects a naive approach of a FL algorithm called FedAvg [21] on the global model with BN layers [19, 17]. When $\{\mu_l(x^*; w), \sigma_l(x^*; w)^2\}_{l=1}^M$ is shared from a client to the server for the update of population statistics in the global model's BN layers, the server as an honest-but-curious adversary would work to add up the batch statistics matching loss term to (1) to ensure a stronger attack.

Then, optimization problem in (1) can be rewritten by considering both assumptions mentioned previously as follows:

$$\arg\min_x \mathcal{L}_{grad}\left(\frac{\partial\mathcal{L}(f_w(x),y^*)}{\partial w}, \frac{\partial\mathcal{L}(f_w(x^*),y^*)}{\partial w}\right) + \alpha_{prior}\mathcal{R}_{prior}(x) + \alpha_{BN}\sum_{l=1}^{M}\mathcal{R}_{BN}((\mu_l,\sigma_l^2),(\mu_l^*,\sigma_l^{*2}))$$
(2)

, where $\mathcal{R}_{BN}$ is the BN statistics matching loss and $\alpha_{BN}$ being its coefficient with $(\mu_l,\sigma_l) = (\mu_l(x;w),\sigma_l^2(x;w))$ and $(\mu_l^*,\sigma_l^*) = (\mu_l(x^*;w),\sigma_l^2(x^*;w))$.

By solving the optimization problem in (2), high resolution images (e.g. ImageNet [4]) with a batch size of up to 40 can be constructed in [26]. However, $f_w$ is only considered for three models: ImageNet pre-trained ResNet18 model, ImageNet pre-trained ResNet50 model, and MOCO v2 [2] pre-trained ResNet50 model fine-tuned with ImageNet. However, there are various choices of $f_w$. Although a broad spectrum of $f_w$ choices is introduced in [6] (e.g., increasing channel size, models with or without skip connection), the authors of the work verified the effect from model variations on single image reconstruction as well as considered the optimization problem of the form (1) rather than (2). Thus, in this paper, we recap how model variations considered in [6] affect reconstruction of multiple images in a batch by solving optimization problem of the form (2) to achieve a better quality of reconstructed samples.

## 3 Re-evaluation of SOTA Gradient Attacks on a Broad Spectrum of Models

Prior works in gradient inversion attacks properly select limited range of models with vulnerability under the proposed attack methods to demonstrate their effectiveness [26, 12, 30, 6]. Therefore, this study aims to re-evaluate state-of-the-art attack methods on a broad spectrum of models. The target of our evaluation is attack methods that can solve the optimization problem of the form (2) assuming that the server as an honest-but-curious attacker desires to reconstruct multiple private images from batch gradients given, which is rarely studied previously. The model variations we considered are twofold: *implicit* and *explicit*.

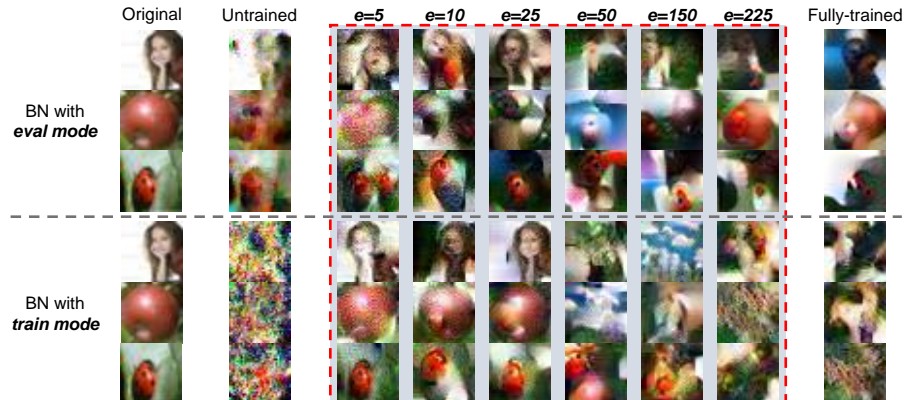

Figure 1: **Visualization of reconstructed images from implicit model variations of ResNet18.** Here $e$ denotes training epochs. Then, "Untrained" means $e = 0$, and "Fully-trained" means $e = 300$ as the ResNet18 model is trained on CIFAR100 training set up to 300 epochs. Reconstructed images in the red dotted line box come from our choices of $e$. Original images (a woman image, an apple image, a beetle image) were randomly sampled from the CIFAR100 validation set.

### 3.1 Implicit model variation: BN modes and training epochs

While *explicit model variation* refers to an architectural change such as increasing channel sizes of the model, as suggested in [6], *implicit model variation* is invisible in the architectural level. However, changes arise internally within the same architecture such as applying different weights with different training epochs or switching the mode of normalization layers (e.g., switching between train and eval modes for BN). This is the first work to introduce the concept of implicit model variation. More

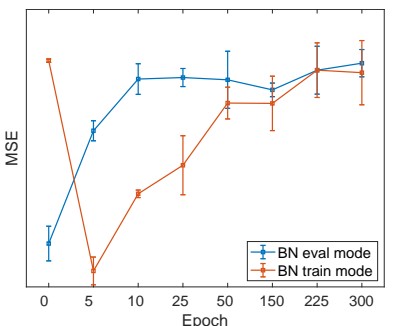 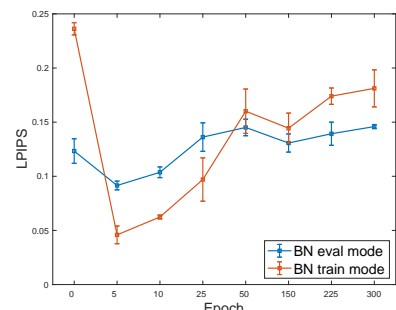

Figure 2: **Plotting the quality of reconstructed samples from implicit model variations of ResNet18 in terms of MSE (↓, left) and LPIPS (↓, right).**

specifically, this is the first time implicit model variation is considered for the evaluation of gradient inversion attacks. Interestingly, we experimentally found that the reconstruction quality ranges in a broad spectrum over *implicit model variations*.

### 3.1.1 BN modes: motivation

State-of-the-art gradient inversion attack methods elevate the quality of reconstructed samples by introducing batch statistics matching loss to the original problem of gradient matching as in (2). Therefore, we adopt a global model with BN layers to realize shared batch statistics in FL. BN layer has two modes of operation: train mode and eval mode [11]. However, recent works have not specified which mode is set for their demonstration while the malicious server, at least as an honest-but-curious attacker, can send a global model with BN layers set to any mode. Therefore, this study considers both BN train mode and BN eval mode for the re-evaluation of SOTA gradient attacks. Our re-evaluation results show that reconstruction results from different BN modes can be significantly different from each other even in terms of the same model weights as in Tab. 1.

| Epoch ($e$) | MSE ↓ | PSNR ↑ | LPIPS ↓ |
|---|---|---|---|
| 0 | **0.8499 ± 0.1996** | **12.8833 ± 1.241** | 0.1233 ± 0.0227 |
| 5 | 1.5033 ± 0.1157 | 10.4366 ± 0.43 | **0.0915 ± 0.0081** |
| 10 | 1.7985 ± 0.1766 | 9.87 ± 0.2749 | 0.1037 ± 0.0099 |
| 25 | 1.8072 ± 0.1042 | 10.02 ± 0.5716 | 0.1362 ± 0.0263 |
| 50 | 1.7941 ± 0.3291 | 9.8666 ± 0.5507 | 0.1451 ± 0.0153 |
| 150 | 1.7361 ± 0.0783 | 10.34 ± 0.3732 | 0.1307 ± 0.0167 |
| 225 | 1.8495 ± 0.2759 | 10.1866 ± 0.4878 | 0.1393 ± 0.0214 |
| 300 | 1.8899 ± 0.1575 | 9.75 ± 0.4313 | 0.1459 ± 0.0038 |

(a) BN with eval mode

| Epoch ($e$) | MSE ↓ | PSNR ↑ | LPIPS ↓ |
|---|---|---|---|
| 0 | 1.9045 ± 0.0195 | 8.9265 ± 0.0665 | 0.2362 ± 0.0113 |
| 5 | **0.6921 ± 0.1601** | **14.9733 ± 0.9168** | **0.0459 ± 0.0164** |
| 10 | 1.1367 ± 0.0434 | 12.5 ± 0.2861 | 0.0624 ± 0.0035 |
| 25 | 1.3015 ± 0.3402 | 11.6733 ± 1.4027 | 0.097 ± 0.04 |
| 50 | 1.66 ± 0.1831 | 10.0433 ± 0.6354 | 0.1601 ± 0.041 |
| 150 | 1.6581 ± 0.3138 | 10.2066 ± 1.1074 | 0.1444 ± 0.0279 |
| 225 | 1.8497 ± 0.315 | 9.49 ± 1.008 | 0.174 ± 0.0151 |
| 300 | 1.8353 ± 0.3703 | 9.4333 ± 0.8832 | 0.1812 ± 0.0342 |

(b) BN with train mode

Table 1: Quantitative comparison between reconstruction results for 50 CIFAR100 images from ResNet18 model with BN set to (a) eval mode and (b) train mode. MSE (↓), PSNR (↑), and LPIPS (↓) are used as evaluation metrics. We highlight the best performance for each column in **bold**.

### 3.1.2 Training epochs: motivation

In a scenario of FL, a client can participate at any time during training. Then, a client can encounter the global model with arbitrary performance. This fact contradicts previous works' experimental setup, where the global model is chosen in a dichotomous manner: an untrained (or initialized) model or a model fully trained on the training set [6, 26]. Therefore, we re-evaluated SOTA inversion attacks on models with a broad spectrum of training epochs. We empirically found that the best reconstruction quality is usually obtained at earlier training epochs.

### 3.1.3 BN modes and training epochs: experimental results

**Setup** We trained a ResNet18 model on CIFAR100 [15] training set for 300 epochs using SGD optimizer with initial learning rate 0.1, momentum 0.9, and learning rate decay 0.1 applied when

$e = 150$ and $e = 225$ for the training epoch $e$. During training, we saved checkpoints of model weights when $e \in \{0, 5, 10, 25, 50, 150, 225, 300\}$ to consider the models from different training epochs. We oversampled model weights before the first learning decay ($0 < e < 150$) to cover the whole set of dynamically changing model weights in the beginning of training. On the other hand, hyperparameters and loss function choices for input reconstruction attacks are borrowed from [10].

**Results** As expected from their difference in batch statistics computation, BN with train mode and BN with eval mode show different reconstruction results both qualitatively (see Fig. 1.) and quantitatively (see Fig. 2 and Tab. 1.). When BN is set to eval mode, partial information (e.g. colors or shapes) is barely leaked in reconstructed images only for the cases $e = 0$ and $e = 5$ as described in Fig. 1 and Fig. 2. On the other hand, for BN with train mode, the quality of reconstructed images were sufficient enough to identify the object in each image only for the cases $e = 5, 10, 25$. Unlike the BN mode set to eval mode, it is remarkable that reconstructed images from BN with train mode in Fig. 1 are noisy images for $e = 0$. For the cases $e \geq 50$, input reconstruction failed for both BN modes and reconstructed images even from the same target gradients look significantly different for different BN modes. However, both BN with train mode and BN with eval mode have similar reconstruction quality in terms of both mean squared error (MSE) and Learned Perceptual Image Patch Similarity (LPIPS) [28] in Fig. 2 and Tab. 1. Therefore, *in the early stage of training, a global model would be privacy threatening with high probability.*

## 3.2 Explicit model variation: skip connection and channel size

Explicit model variations involve *change in architecture level* like removing skip connections in residual blocks or increasing the number of channels in convolution module, which are the kinds considered in previous works but on single image reconstruction. Therefore, we re-explore the effect of skip connection and channel size on the model's vulnerability against gradient inversion attack but in the context of batch reconstruction. Skip connection helps information flow both forward and backward through the network, thus input reconstruction is expected to be easier for residual networks but harder for models without skip connection [9]. On the other hand, increasing channel size implies increasing dimension of gradients, which is the capacity of gradients to store information. Therefore, we expect that more information about input would be compressed in gradients when the number of channels increase.

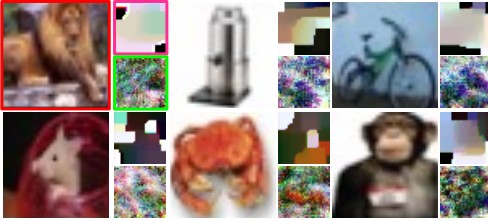

Figure 3: **Visualization of reconstructed images from ConvNet on CIFAR100.** In each image block, the images at the positions of the red, pink, and green borders denote the original image, the reconstruction with BN (*eval mode*), and the reconstruction with BN (*train mode*), respectively. Original images were randomly sampled from CIFAR100 validation set.

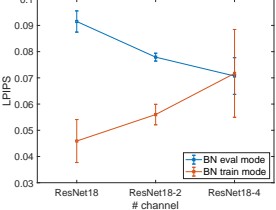

Figure 4: **Plotting the best reconstruction quality in terms of LPIPS (↓) among model variations through training epochs for ResNet18, ResNet18-2, ResNet18-4 models with BN *eval* (orange) and *train* (blue) modes.** $e = 5$ or $e = 10$ usually result in the best reconstruction quality. As channel size increases, the reconstruction quality increases for BN *eval* but decreases for BN *train*.

### 3.2.1 Skip connection and channel size: experimental results

**Setup** Instead of ResNet18, we trained a ConvNet model, ResNet18-2 model, and ResNet18-4 model for explicit model variations. ConvNet, which is introduced in [6] for the first time, is a convolutional neural network without skip connection and ResNet18-2 and ResNet18-4 being ResNet with channel size doubled and quadrupled, respectively. Note that we apply implicit variations considered in the Sec. 3.1 to the models. Training conditions and hyperparameters for both model training and attack methods are kept the same with the setup in the previous section.

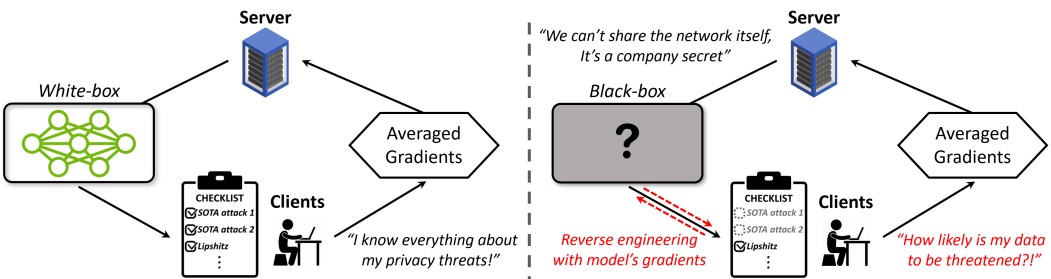

Figure 5: **White-box (left) and black-box (right) FL settings.**

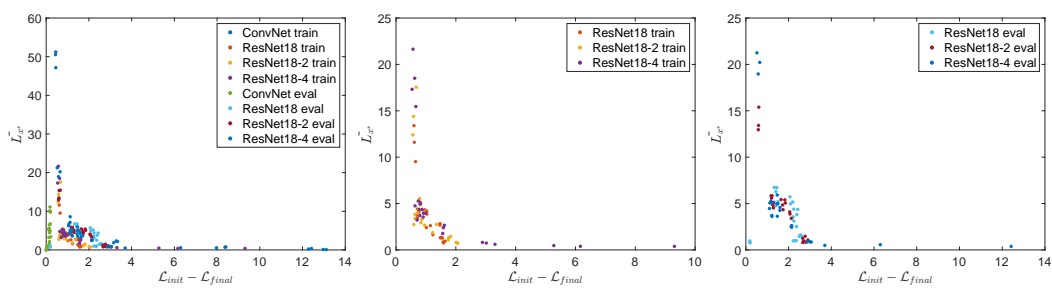

(a) All models ($r_s = -0.78$)    (b) ResNets, BN *train* ($r_s = -0.87$) (c) ResNets, BN *eval* ($r_s = -0.66$)

Figure 6: **Proposed measure $\tilde{L_{x^*}}$ is approximately a monotonic decreasing function with respect to $\mathcal{L}_{init} - \mathcal{L}_{final}$, the difference between initial ($\mathcal{L}_{init}$) and final losses ($\mathcal{L}_{final}$) among (a) all models, (b) ResNet models with BN *train* mode, and (c) ResNet models with BN *eval* mode considered in Sec. 3.**

**Results**  Reconstructed images from ConvNet models with the best quality, in terms of LPIPS, are listed in Fig. 3. For ConvNet models, reconstructed images, even with the best quality, are far from original images visually due to severe artifacts. Therefore, as expected from the role of skip connection in residual networks, a network without skip connection like ConvNet seems to be robust against input recovery attacks. Then, ConvNet models would be considered as global model candidates for privacy protection in FL despite of their worse performance than that of residual networks.

By contrast, the best averaged reconstruction results among the sampled training epochs $e \in \{0, 5, 10, 25, 50, 150, 225, 300\}$ are plotted in Fig. 4 for ResNet18, ResNet18-2, and ResNet18-4 models with varied BN modes. When BN is set to the eval mode, the reconstruction quality increases as the number of channels increases as expected. However, the reconstruction quality worsens as the number of channels increases for BN set to the train mode, which breaks the belief from previous works that increasing channel size makes input recovery attack easier [6, 30]. However, the reconstruction quality obtained with BN train mode is better than that with BN eval mode for all models considered except ResNet18-4, where their LPIPS range overlaps, implying that BN train mode is vulnerable against input recovery attacks than BN eval mode. The quantitative results for ConvNet, ResNet18-2, and ResNet18-4 are provided in Appendix A1.

## 4 Lipschitz Smoothness for Client-Side Privacy Leakage Detection

For privacy-preserving FL, choosing global model robust against any server-side input reconstruction attack method would be important. At the least, global model should be robust against well-known SOTA gradient inversion attack methods to alleviate clients' anxiety about any potential leakage from gradient sharing with the server. If clients can access the global model with the same level of a central server (*white-box*), applying SOTA attack methods directly to the global model with private images would be the best way for assessing whether or not the global model presents a risk to the client's privacy. However, in general, global model information would be opaque to clients due to company

secrets. As clients should communicate with the server via locally computed gradients, we suppose the *black-box* setting, where model gradients are given to clients as minimal information of the global model. Therefore, we provide a helpful measure for developing the system for clients to examine whether the given global model is safe in terms of privacy by using gradients computed with their self-controlled inputs. Note that *white-box* and *black-box* are described in Fig. 5.

## 4.1 Angular Lipschitz smoothness: motivation

If a function $f : \mathbb{R}^n \to \mathbb{R}^m$ is Lipschitz smooth (or the derivative of $f$ is Lipschitz continuous) with constant $L$, then the following holds: $||\nabla f(x) - \nabla f(y)|| \le L||x - y|| \; \forall x, y \in \mathbb{R}^n$. The concept of Lipschitz smoothness or Lipschitz continuity is frequently employed to prove convergence theorem of gradient descent methods for optimization [24, 7, 27, 20, 18, 1]. This study employs the concept of Lipschitz smoothness to prove the following theorem in the context of gradient matching problem.

**Theorem 1** (Monotonic decreasing loss function). Suppose $\nabla_w \mathcal{L}(f(x), y)$ is Lipschitz continuous with respect to $x$ with constant $L$ and $\mathcal{L}_{grad}^x = ||\nabla_w \mathcal{L}(f(x), y) - g^*||_2^2$ is given as a gradient matching loss. Then, when gradient descent $\triangle x$ is applied with step size $\mu > 0$ and $L > \epsilon$ for some $\epsilon > 0$, the following holds:

$$\mathcal{L}_{grad}^{x+\triangle x} \le \mathcal{L}_{grad}^x - \frac{1}{L^2}||\frac{\partial \mathcal{L}_{grad}^x}{\partial x}||_2^2. \tag{3}$$

Inequality (3) implies that gradient matching loss strictly decreases as the gradient descent steps unless the gradient term $\frac{\partial \mathcal{L}_{grad}^x}{\partial x}$ is zero (i.e. gradient matching loss already converges). Furthermore, a gradient descent with a small $L$ (or large $\frac{1}{L^2}$) can accelerate the convergence of gradient matching optimization but with the premise that $L > \epsilon$ for $\epsilon > 0$. This premise is required to ensure the first-order Taylor approximation for $\nabla_w \mathcal{L}(f(x + \triangle x), y)$ in the proof in Appendix A2. Therefore, in a particular range of $L$ (i.e., $L > \epsilon$), we hypothesize that a global model with smaller $L$ experiences a sharper loss drop in gradient matching optimization. *We empirically found that $L$ is not too small for most models, thus meeting the premise in reality*.

For the empirical verification of the hypothesis in the context of input reconstruction, we desire to compute Lipschitz smoothness constant locally around $x^*$, $L_{x^*, \epsilon} = \sup_{||x - x^*|| < \epsilon, x \ne x^*} \frac{||\nabla_w \mathcal{L}(f(x), y) - \nabla_w \mathcal{L}(f(x^*), y)||}{||x - x^*||}$, with small $\epsilon$, for the models considered in Sections 3.1 and 3.2. Recent works on computing precise upper bound of $L$ only focus on multi-layer perceptrons (MLP) due to the difficulty of computing $L$ for normalization layers or residual layers. Therefore, $L_{x^*, \epsilon}$ is estimated as $\tilde{L_{x^*}} = \max_{n \ne \mathbf{0}} \frac{||\nabla_w \mathcal{L}(f(x^*+n), y) - \nabla_w \mathcal{L}(f(x^*), y)||}{||n||}$ by sampling 1,000 noises ($n$) from the Gaussian distribution $\mathcal{N}(0, 0.001^2)$ in our experiments.

However, $\tilde{L_{x^*}}$ can be any nonnegative value by scaling loss function $\mathcal{L}$. If $\mathcal{L}$ is scaled by nonnegative scalar $k$, then $\tilde{L_{x^*}}$ is scaled by $k$ too, allowing $\tilde{L_{x^*}}$ to be manipulated by the server using simple loss scaling. Therefore, inspired by the cosine similarity loss function, which is scale-invariant, we propose *the angular Lipschitz constant* $\tilde{L_{x^*}^{cos}} = \max_{n \ne \mathbf{0}} \frac{1 - cs(\nabla_w \mathcal{L}(f(x^*+n), y), \nabla_w \mathcal{L}(f(x^*), y))}{1 - cs(x^*, x^*+n)}$ ($cs$ being the cosine similarity loss) as a loss scaling-invariant alternative to $\tilde{L_{x^*}}$. We find hat $\tilde{L_{x^*}^{cos}}$ shows a strong monotonic correlation with the quality of reconstructed samples, demonstrating the potential of $\tilde{L_{x^*}^{cos}}$ to be imperative for client-side defense methods.

## 4.2 Angular Lipschitz smoothness: experimental results

We computed $\tilde{L_{x^*}}$ and the attacker's loss drop $\mathcal{L}_{init} - \mathcal{L}_{final}$ ($\mathcal{L}_{init}$ and $\mathcal{L}_{final}$ being the initial and final losses, respectively) for the models and input batches considered in Sec. 3 (Fig. 6a). We also quantified their correlation using the Spearman's rank correlation coefficient $r_s$, which quantifies how two variables are in a monotonic relationship. $r_s = 1$ ($r_s = -1$) means that one variable is a completely monotonic increasing (decreasing) function with respect to the other one. Then, $\tilde{L_{x^*}}$ is almost a monotonic decreasing function with respect to $\mathcal{L}_{init} - \mathcal{L}_{final}$ with $r_s = -0.78$, thus validating our hypothesis. For ResNet models with BN *train* (Fig. 6b), $\tilde{L_{x^*}}$ and $\mathcal{L}_{init} - \mathcal{L}_{final}$ show a stronger monotonic correlation than that for ResNet models with BN *eval* (Fig. 6c) with

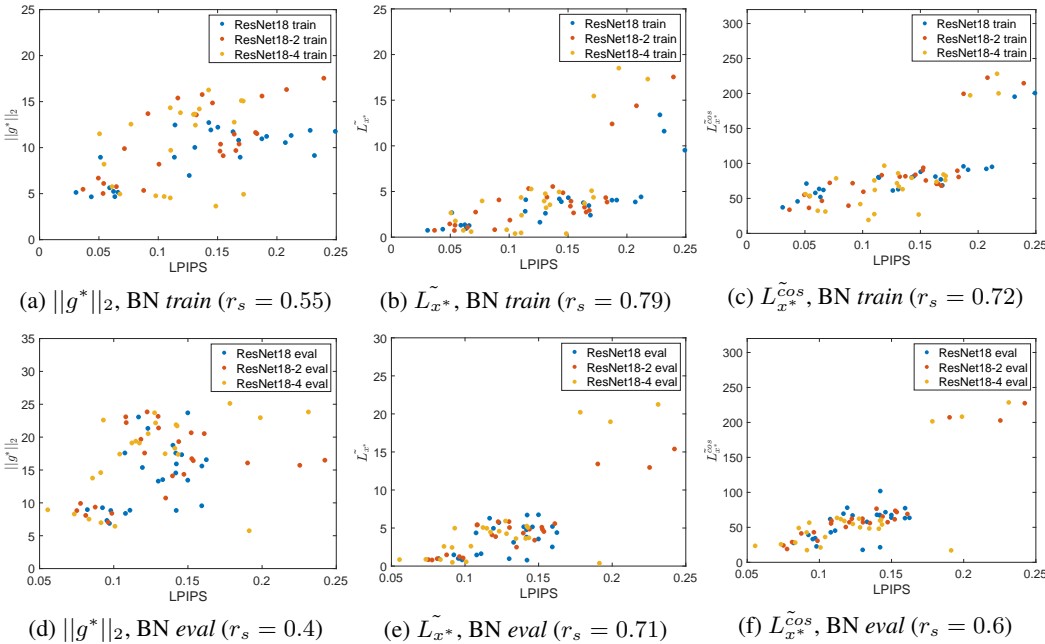

(a) $||g^*||_2$, BN *train* ($r_s = 0.55$)  (b) $\tilde{L_{x^*}}$, BN *train* ($r_s = 0.79$)  (c) $L_{x^*}^{\tilde{cos}}$, BN *train* ($r_s = 0.72$)

(d) $||g^*||_2$, BN *eval* ($r_s = 0.4$)  (e) $\tilde{L_{x^*}}$, BN *eval* ($r_s = 0.71$)  (f) $L_{x^*}^{\tilde{cos}}$, BN *eval* ($r_s = 0.6$)

Figure 7: **Comparison of $||g^*||_2$, $\tilde{L_{x^*}}$, and $L_{x^*}^{\tilde{cos}}$ in terms of the correlation between LPIPS of reconstructed samples for ResNet models with BN *train* (top) and BN *eval* (bottom)**

$r_s = -0.85$. As in Tab. 1 and Fig. 1, reconstructed samples are closer to their original images in BN *train mode*, thus $\tilde{L_{x^*}}$, which is computed around the ground truth $x^*$, seems to fit more to BN *train* while $L$ should be estimated around the solution from the attack method rather than $x^*$ for the case of BN *eval*. However, clients cannot access to the solution from the the attack method in the *black-box* setting. The plot of $\tilde{L_{x^*}}$ and $\mathcal{L}_{init} - \mathcal{L}_{final}$ for the ConvNet models is provided in Appendix A3.

## 5 Limitations and Future Work

Our hypothesis can be extended to the correlation between Lipschitz constant and the quality of reconstructed samples, rather than loss drop. Zero gradient matching loss does not mean complete recovery of original images due to the existence of *twin data* [30], two different data input with identical model gradients. However, we empirically found that both $\tilde{L_{x^*}}$ and $L_{x^*}^{\tilde{cos}}$ show positive monotonic correlations with the quality of reconstructed samples, in terms of LPIPS (lower value is better) (Fig. 7). In particular, they beat the baseline measure, the norm of given gradients ($||g^*||_2$), which was implicitly believed to be the amount of information within the gradients in previous works, by a wide margin, in terms of $r_s$. Therefore, we expect $\tilde{L_{x^*}}$ and $L_{x^*}^{\tilde{cos}}$ to be the key factors for developing future client-side defense strategies.

## 6 Conclusions

Here, we re-evaluated the SOTA attack method on a broad spectrum of models in the context of batch reconstruction, which is rarely studied in previous works. We considered model variations of two types: *implicit*, which changes in model weights or BN modes within the same architecture, and *explicit*, with changes in architecture. The re-evaluation results indicate that the quality of the reconstruction attack varies depending on the implicit or explicit model changes. Therefore, inspired by our theorem related to the convergence of gradient matching optimization and scale-invariance of the cosine similarity loss function, we propose an explainable and predictive measure for privacy leakage, an angular Lipschitz constant $L^{cos}$, which is invariant to trivial loss scaling attacks from malicious servers. We empirically find that $L^{cos}$ shows a strong monotonic correlation with the quality of reconstructed samples, thus expecting the potential of $L^{cos}$ to be a key factor for clients' defense strategies in a *black-box* setting, where only model gradients are given as minimal information about the global model to clients.

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
