# OpenReview forum: "Foreseeing Privacy Threats from Gradient Inversion Through the Lens of Angular Lipschitz Smoothness"
_NeurIPS.cc/2022/Conference — NeurIPS 2022 Submitted_

### Official Review · Reviewer_F6kS · 2022-07-08

**Rating:** 3
**Confidence:** 4
**Soundness:** 2 fair
**Presentation:** 2 fair
**Contribution:** 2 fair

**Summary:**

This paper studied the gradient inversion problem where an honest-but-curious server aims to reveal clients’ private data from model weights and gradients shared by clients in federated learning (FL). The paper first evaluated SOTA gradient inversion algorithms (mainly [6] with an additional BN-related loss term), under implicit model variations (different BN modes and training epochs with the same model architecture) and explicit variations (different model architectures). The evaluation showed that the reconstruction results vary across model variations both qualitatively and quantitively. The paper then proposed a measure of angular Lipschitz constant to indicate the outcome of reconstruction, which was shown to be stronger correlated with LPIPS and attack loss drop than gradient norm.

**Questions:**

- The paper mentioned about knowledge of target labels and batch size in previous work. What are the related settings in experiments of this paper? Batch size may be an important factor for observing the effect of BN modes on the performance.
- It is stated in Line 168 that the server “can send a global model with BN layers set to any mode”. But BN modes are about whether to use running mean/variance or current batch’s mean/variance in BN layers, and the computation is done on clients. How can an honest-but-curious server alter the BN mode without being noticed?
- The computation of the angular Lipschitz constant (defined in Line 277) is done through randomly sampling from Gaussian. Can this measure be manipulated by scaling the input x^*, for fixed variance of Gaussian noises? Is the input x^* normalized to compute the measure?
- In the theoretical result, the loss is guaranteed to monotonically decrease only when the step size \mu is small enough (optimally =1/(2L^2)). Are the learning rates sufficiently small in experiments?
- What is \delta in Lines 458, 465 and 466?
- BN statistics are used in two places in the inversion process, one in equation (2) as a loss term, and another in the BN layers of the networks (in train/eval modes). Which one affects the performance of inversion algorithms more significantly?


**Limitations:**

The paper discussed limitations in Section 5.

**Strengths And Weaknesses:**

Strengths:
- The paper addressed an interesting problem of evaluating gradient inversion algorithms for different models. A systematic experimental study may help gain insights about the mechanism of gradient inversion.
- The paper made a good summarization on the optimization-based gradient inversion approach and related prior art in Section 2.
- The paper showed some interesting results regarding the BN train mode. Firstly, the inversion achieves the best quantitative performance at early training stages (e=5). Secondly, the inversion achieves best results for ResNet18 than for ResNet18-2 and ResNet18-4, showing that wider networks are not necessarily easier to attack.

Weaknesses:
- The main contribution of this paper seems ambiguous. The title and abstract suggest that the paper mainly proposes a new measure, angular Lipschitz constant, for assessing inversion attacks, but a large body of the paper focuses on re-evaluation of existing methods over model variations. While the re-evaluation is valuable, it should be more clearly stated as primary results or just motivational study.
- The scope of re-evaluation of SOTA algorithms in this paper overlaps with [10], which focused on evaluating gradient inversion attacks and defenses. In particular, [10] evaluated the algorithm in [6] with BN loss term (as in equation (2) of this paper), and considered different settings on knowledge of BN statistics. What is the difference between the BN settings of this paper and [10]? A detailed discussion and comparison with [10] would be beneficial.
- The re-evaluation of SOTA algorithms was performed on limited variations. If the empirical study is positioned as the primary contribution, a wider range of models may be considered, such as initialization schemes (pre-trained or random) and random seeds for implicit variations, and more architectures beyond ResNet18 with different widths for explicit variations.
- The paper only considered CIFAR100 images in experiments, but not higher-resolution (e.g. ImageNet) images.
- The paper motivates the proposed angular Lipschitz constant measure with the case that clients may only have access to gradients provided by the server. However, wouldn’t the client need white-box access to compute gradients using private local data for model training in FL? If the client only has black-box access, what is the interplay between model training, gradient inversion, and risk assessing (Section 4)? And how does the client compute the measure (in Line 277) through random sampling?

---

> ### Author Response · Authors · 2022-08-02
> **Rebuttal for R4's comments**
>
> # W1 : The main contribution of our work is ambiguous
>
> The first part of our paper is positioned as both primary results and motivational study. We found some counterintuitive results which break the belief of previous work[10] (e.g. the relationship between channel size and reconstruction quality). In addition, the results motivate us to propose the measure that can explain themselves. Please see C2 for more details.
>
> # W2 : The novelty of our work regarding BN modes
>
> Please see C2 for more details. We focus on gradient computation rather than statistics variation regarding BN modes. We always compute exact batch statistics (No running statistics shared for BN eval mode).
>
> # W3 : Limited model variations
>
> Our model variations in the original version are still novel. Please see C2 for more details. In addition, the suggestion about the initialization scheme by R4 intrigues us and we conducted experiments validating our measure on self-supervised models. We also added the results on ImageNet single image recovery, but on additional architectures, such as MobilieNet v2 and several ShuffleNet variations. The results are included in the appendix. Please see C3 for more details.
>
>
> # W4 : Higher resolution images
>
> We added the results on ImageNet single image recovery. Please see C3 for more details. Our proposed measure acts as an upper bound of reconstruction quality solidly in diverse settings.
>
> # W5 : Black-box setting
>
> Please see C1 for more details.
>
> # Q1 : Target labels and batch size
>
> We assume correct target labels are known to the attacker for stronger attacks and batch size is set to 16 as in [7].
>
> # Q2 : How can the server alter BN mode?
>
> In black-box setting, the model is protected, thus clients have a restricted access to the models, thus BN mode is set by the server, rather than clients. Please see C1 for more details.
>
> # Q3 : Manipulation of angular Lipschitz constant through scaling input images
>
> Only clients can manipulate input images, thus there is no need for clients to manipulate angular Lipschitz constant. Instead, we scale the noise instead of input depending on the dataset. For CIFAR, noise has standard deviation as 0.001 while noise has standard deviation as 0.01 for ImageNet.
>
> # Q4 : sufficiently small learning rate
>
> We select an optimal initial learning rate among 0.1, 0.01, and 0.001 by running each of them. Because learning rate decay 0.1 occurs twice during training, the final learning rate would be 0.001, 0.0001, and 0.00001, which might cover sufficient range of optimal earning rates.
>
> # Q5 : \delta in Lines 458, 465, and 466
>
> \delta is a sufficiently small scalar that guarantees the Taylor’s first order approximation in our proof of Theorem 1.
>
> # Q6 : The significance of BN statistics in network and BN stats matching loss term
>
> We found that the trend about the relationship between our measure and reconstruction quality is managed even when BN stats matching loss term is excluded from optimization. Please see Figures 13-14 and Figures 15-16 of the appendix (supplementary material) for comparing both cases.

---

> ### Author Response · Authors · 2022-08-08
> **R4 - The discussion period closes in about 24 hours**
>
> Today is the end of the discussion deadline. Could you please go over our rebuttal and check the responses? We believe that we have addressed all your concerns and that including these discussions will further strengthen our paper. We hope you reflect this in your final review and the score. We thank you again for your time and efforts in reviewing our paper.
>
> Thank you,
> Authors.

---

> > ### Comment · Reviewer_F6kS · 2022-08-09
> > **Thank you for the reminder**
> >
> > I have read the above response and discussion, while I still have some concerns. Firstly, the black-box setting seems largely different from conventional FL and may need discussion separately. In the black-box setting, the clients have no idea about how their data are used and may not be able to assess the privacy threat against the server, which implicitly weakens the defence capability. For instance, an honest-but-curious server is supposed to passively infer clients' information while all parties know and follow the (white-box) federated training paradigm. But in the black-box setting, as the paper mentioned, the server can alter BN modes arbitrarily, without being noticed by clients. And the server can simply take more aggressive actions to extract information that are covered by the black-box. Secondly, substantial experiment results are added in revision. While I appreciate these results, they also make the paper lean towards an empirical evaluation study. I think it may be important to sort out what key insights/contribution are drawn from these results.

---

> > > ### Author Response · Authors · 2022-08-09
> > > **Response to the remaining concerns of R4**
> > >
> > > Thanks for your time and efforts spent for discussion.
> > >
> > > First, our intention of mentioning 'black-box' setting in our paper is to claim that clients require at least true gradient information
> > > to identify robust FL models against optimization based gradient inversion attacks. The basic federated learning assumes that
> > > clients faithfully send true gradient information to the server while some malicious clients can arbitrarily send wrong gradient information
> > > to harm the FL model, which is the unwanted case. Similarly, our conceptual 'black-box' setting assumes that the client program
> > > made by the server faithfully outputs true gradient information to let clients predict the vulnerabilities against gradient inversion through
> > > our proposed measure. Obviously, the malicious behavior by the server mentioned by R4 encourages the future work for
> > > more reliable protocol between clients and the server. Also, please note that our proposed measure can act as an efficient predictor
> > > for reconstruction quality of gradient inversion attacks even in 'white-box' setting.
> > >
> > > Second, our additional experiments on broader settings are about experimental validation of our theoretically derived measure, angular
> > > Lipschitz smoothness. Please check that the correlation between our proposed measure and reconstruction quality is consistently shown
> > > on even high resolution images with diverse models including MobileNetV2, ShuffleNetV2 variations, and several self-supervised Resnet50 models. Additional experiments are not about just saying that some models are robust and some models are vulnerable against model inversion attacks, but about saying that our proposed measure can be applicable on broader settings.
> > >
> > > **We hope you reflect this in your final review and the score. We thank you again for your time and efforts in reviewing our paper.**

---

> ### Author Response · Authors · 2022-08-09
> **R4 - The discussion period closes in about 3 hours**
>
> Today is the end of the discussion deadline. Could you please check our clarification on 'black-box' setting and meaning of additional experiments as an experimental validation of our measure in additional responses? We believe that we have addressed all your concerns and that including these discussions will further strengthen our paper. **We hope you reflect this in your final review and the score. We thank you again for your time and efforts in reviewing our paper.**
>
> Thank you,
> Authors.

---

### Official Review · Reviewer_vsQD · 2022-07-11

**Rating:** 2
**Confidence:** 4
**Soundness:** 1 poor
**Presentation:** 3 good
**Contribution:** 1 poor

**Summary:**

The submission "Foreseeing Privacy Threats from Gradient Inversion Through the Lens of Angular Lipschitz Smoothness" is concerned with gradient inversion attacks in federated learning. The first part of this submission contains additional results and ablation studies about the success conditions of such attacks. The second part proposes a new measure of vulnerability that is claimed to be a key to client-sided defenses.

**Questions:**

Maybe I misunderstood the intent of the submission concerning the black-box setting. I would be glad about a correction or additional insights why this is a meaningful setting.

**Limitations:**

The submission discussses some limitations, however limitations of the proposed measure are not discussed. There is not enough evidence presented whether this really would be a key factor in future defenses.

**Strengths And Weaknesses:**

I, unfortunately,  have several points of criticism with this work, which I will list out below:

* This submission analyzes gradient inversion attacks as developed around 2019/2020, but of the analysis given in this submission has been included in other work also evaluating these attacks since. Questions brought up concerning batch normalization have been  already been analyzed in works such as Huang et al., "Evaluating Gradient Inversion Attacks and Defenses
in Federated Learning", from last year's NeurIPS. Solutions to the normalization question have then been proposed in Hatamizadeh et al., "Do Gradient Inversion Attacks Make Federated Learning Unsafe?". Whether labels are required was also discussed in Huang et al., and work such as Wainakh et al, "User Label Leakage from Gradients in Federated Learning" have made strides towards showing that label knowledge is not a requirement for a succesful attack. The impact of training state in recovery success has been discussed several times, from Zhu et al, "Deep Leakage from Gradients", Geiping et al., "Inverting Gradients - How easy is it to break privacy in Federated Learning" and also appears in follow-up work such as Wei et al., "A Framework for Evaluating Gradient Leakage Attacks in Federated Learning".

* The submission states that choices in Zhu et al., were based on the unavailability of automatic differentiation. Yet, AD was certainly already developed and present in 2018! These paragraphs in the related work are incorrect, and it is unclear what "computed the loss in closed form" here should mean. The existence of closed form solutions (which Zhu et al. do not employ, as they also backpropagate gradients using AD) is unrelated to the smoothness of the loss function. Zhu et al. use smooth activations because their optimization strategy is based on an L-BFGS solver that nominally requires well-defined higher-order derivatives. Later works use adaptive first-order optimizer and hence drop this requirement.

* I am doubtful about the proposed black-box setting. The submission proposes a black-box setting in which the global model parameters are hidden from the user, but the user application requires these parameters to compute the update gradient. The parameters have to be send to the user device, and the user has full control over their device, so the users necessarily have access to model parameters as well.

* Further, for a user to use known inversion attacks to gauge their loss of privacy is an inherently unsafe proposal. Even if a user uses these attacks themselves and finds that they cannot reconstruct their data, this is no guarantee of privacy! These attacks can only be used to prove vulnerabilty, never to ascertain safety.

* The introduction mentions evaluation of additional models. However in the experimental section I find only the ConvNet and ResNet variations used in previous work?

* Minor: The cited work by Bauschke, Bolte and Teboulle is one of their few work where Lipschitz smoothness of the gradient in the classical sense is not actually required, in contrast to how this work is cited here.

* I like the general idea of the section concerning angular Lipschitz smoothness, but the concept makes up such a small part of the submission that it is difficult to quantify its effectiveness. The submission shows some correlation with attack success if measured in LPIPS, but the correlation is not overly strong and it is unclear whether this is a strong argument for robustness. LPIPS scores are only be a limited measure of safety. The main question here would be whether some threshold of angular Lipschity smoothness would imply that no reconstruction succeeds. The measure discussed in Yin et al., "See through gradients" might be helpful here. Further it would be necessary to discuss whether this measure is robust to adaptive attacks, meaning whether this proposed measure can be "gamed" by the attacker in some way. For example, angular smoothness might correlate with gradient obfuscation which would undermine its effectiveness.


* From a more general vantage point, I do think that this submission makes a categorical mistake about the nature of safety. The submission shows scenarios where the (even unmodified) baseline attack works more or less well and concludes from this that the attack should be evaluated in more scenarios, but this misses the asymetric nature of safety research. It is not necessary for a single attack to work optimally in all scenarios, in each scenario the question is whether an attack could exist that breaks privacy. Will small modifications as discussed in this work, categorically defeat all attacks and prove a reliable defense in the future? Will angular smoothness defend against adaptive attacks that circumvent gradient matching losses?

---

> ### Author Response · Authors · 2022-08-02
> **Rebuttal for R3's comments**
>
> # W1 : The novelty of our work
> The focus of our work is generally different from that of previous works. Please see C2 among common comments for understanding our novelty. When it comes to label information, we assume that the attacker knows the correct labels. As suggested by R3, we know that there are some attacks inferring label information from gradients, but we just let an attacker know correct labels without such a trick to focus on stronger image recovery attacks.
>
> # W2 : The emergence of automatic differentiation (AD) precedes the discovery of deep leakage from gradients (DLG)
>
> Thanks for fixing our wrong information in related work. We currently understand that DLG requires architectural modification to use L-BFGS solver. We will revise this part soon.
>
> # W3 : Black-box setting
>
> Black-box setting is clarified in the common comment C1. Our proposed measure is still meaningful as an efficient indicator for the result of known gradient inversion attacks even in white-box settings. Please consider our theoretical results and updated extensive experiments showing the role of our measure as an upper bound of reconstruction quality across high-resolution dataset and diverse models.
>
> # W4 : Non-vulnerability against known inversion attacks are not sufficient for the safety
>
> We used the term ‘safe in terms of privacy’ once when we introduced our measure in the main paper. But we intended to mean that our measure can predict vulnerability of given models under known gradient matching attacks. Please consider that we never used the term such as ‘privacy-preserving’ in our paper, but used the term 'privacy-threatening’. On the other hand, non-vulnerability under known gradient inversion attacks is necessary for the safety, thus practically important. Note that defense strategies are usually evaluated by checking vulnerabilities under known gradient inversion attacks [6,7].
>
>
> # W5 : The novelty of model choices
>
> We considered the same architecture variations as in [10]. However, we consider novel models with checkpoints from different training epochs or different BN modes (train or eval), which was rarely studied in previous works. To give the readers a good insight from our work, we conducted additional experiments for single image recovery of ImageNet on memory-friendly architectures (Mobilenet v2, ShuffleNet variations) and self-supervised models, which are presented in the appendix.
>
> # W6 : The citation of the work regarding Lipschitz smoothness
>
> We will fix this minor issue in the revision.
>
> # W7 : The effectiveness of our angular Lipschitz smoothness and adaptive attack gamed by the server
>
> In our additional experiments, our proposed measure seems to be an upper bound of reconstruction quality from gradient inversion attacks. In other words, as R3 mentioned, there exists some low threshold $t$ so that reconstruction fails with high probability when Lipschitz smoothness exceeds $t$. Although this phenomenon was experimentally verified, this trend was solid even on ImageNet, self-supervised models, and memory-friendly architectures, showing the practical effectiveness of our measure.
>
> We already considered a game by the server in our main paper. The prototype of our measure was just Lipschitz smoothness. However, if loss function is properly scaled by the server, Lipschitz smoothness can have any nonnegative value and this is why we introduce angular Lipschitz smoothness, which is invariant to loss scaling due to cosine similarity term. Note that we proposed our measure as a result of considering attackers’ minds. We plan to consider whether an adaptive attack for our measure is possible or not in future work.
>
>
>
> # W8 : Asymmetric nature of safety research
>
> We understand that an attack method does not need to be successful for all scenarios, but knowing failure cases of attack methods would be meaningful practically. Then, why are the defense strategies experimentally demonstrated through the non-vulnerability against known gradient inversion attacks [6, 7]?  We never mentioned that our measure can predict threats from all attacks including even the attack that circumvents gradient matching losses. In our work, we tried to understand the nature of known gradient inversion attacks using our proposed measure, both theoretically and experimentally. Although this work is only about gradient matching based attacks, understanding the nature of major attacks would be beneficial.

---

> > ### Comment · Reviewer_vsQD · 2022-08-04
> > **Thanks**
> >
> > Thank you for formulating this extensive response.
> >
> > >  Then, why are the defense strategies experimentally demonstrated through the non-vulnerability against known gradient inversion attacks [6, 7]?
> >
> > Defense strategies that are only verified against known attacks are vulnerable to novel attacks and are incomplete without strong adaptive attack evaluations. For example, reference [6] was subsequently broken in Carlini et al., "An Attack on InstaHide: Is Private Learning Possible with Instance Encoding?"

---

> > > ### Author Response · Authors · 2022-08-05
> > > **Adaptive attack for defense methods**
> > >
> > > We agree that defense papers can be perfect by demonstrating them against adaptive attacks. However, our paper is about proposing a measure that correlates with optimization-based gradient inversion attacks, not about a defense strategy.
> > >
> > > Whenever we use the word 'defense' in our paper, we try to express our hope that this measure can be used as an ingredient for a defense method, not the assertion that our measure itself is a defense method. If there is a defense method using our measure, we will keep in mind that an adaptive attack should follow to demonstrate its effectiveness thanks to R3's discussion.
> > >
> > > R3’s comments would be helpful for constructing the detailed design of ‘black-box’ setting and defense methods using our measure in the future. However, our work is currently focused on proposing a measure that correlates with an upper bound of gradient inversion attack quality and our measure is still meaningful as an efficient predictor of gradient inversion attacks in white-box setting.
> > >
> > > Also, please re-consider our theoretical validation and experimental validation across various datasets including ImageNet, and various models including memory-friendly architectures (Mobilenet V2, ShuffleNet V2 variations, VGG19) and even self-supervised models. We hope you reflect this in your final review and score. We are grateful for your time and efforts in reviewing our paper and having discussions that strengthen our paper.

---

> > > > ### Comment · Reviewer_vsQD · 2022-08-09
> > > > **Final Considerations**
> > > >
> > > > I have read all comments presented in reponse to this review and have decided to keep my score. I hope this submission could be rewritten from the ground up to reflect the large amount of feedback and various references provided by the other reviewers and me during the discussion period.

---

> ### Author Response · Authors · 2022-08-08
> **R3 - The discussion period closes in about 24 hours**
>
> We are grateful for your time and efforts spent during discussion period.
>
> First of all, thanks for your comments on the possibility of encrypted gradients in 'black-box' setting.
> In this conceptual setting, we assumed no encryption for gradients to allow clients to compute L-cos locally.
> However, more solid protocols between clients and server are required to tackle 'encrypted gradients' case.
> We hope our paper to open discussions on the design of such protocols.
> Please note that our conceptual setting is motivated by enterprises' preference on closed-source technology and
> our measure is still meaningful as a predictor for gradient inversion attack even in white-box setting.
>
> Second, implicit model variations considred in our paper are novel compared to those in previous works.
> For example, Huang et al. only considers trained ResNet 18 models only and their work is focused on the effect of
> BN statistics with different exactness levels (BN_exact, BN_proxy, BN_infer) on reconstruction quality.
> On the other hand, we focused on how gradient computation differed by BN modes affects reconstruction quality. Note
> that we always use exact batch statistics for both BN eval and train mode to consider stronger attacks. In addition, a fully trained
> ResNet 18 model is just one of the models considered in our work, thus our results are more extensive.
>
> Third, our work is focused on proposing a measure that correlates with reconstruction quality, not proposing a defensive strategy.
> Because gradient matching based attacks are in majority, we intended to devise a measure that correlates with the reconstruction quality
> of such attacks. If there is a defensive strategy using our measure, we surely agree that the defense method should be demonstrated
> under an adaptive attack for completion.
>
>  We believe that we have addressed most of your concerns with more experimental validation of our measure. Please re-consider
> extensive experimental results during rebuttal and theoretical contribution in the original submission. **We hope you reflect this in your final review and the score.**

---

> ### Author Response · Authors · 2022-08-09
> **R3 - The discussion period closes in about 3 hours**
>
> Today is the end of the discussion deadline. Could you please check the additional responses? We believe that we have addressed all your concerns and that including these discussions will further strengthen our paper. **We hope you reflect this in your final review and the score. We thank you again for your time and efforts in reviewing our paper.**
>
> Thank you,
> Authors.

---

### Official Review · Reviewer_9iUp · 2022-07-11

**Rating:** 6
**Confidence:** 4
**Soundness:** 3 good
**Presentation:** 3 good
**Contribution:** 3 good

**Summary:**

This paper works on the privacy attacks in the black-box FL setting, specifically using the gradient inversion methods. The contribution is on the image tasks and empirically solid. Different aspects of models are systematically evaluated to identity the vulnerability to privacy attacks. The authors also propose a new measure, the angular Lipschitz constant, to measure the privacy risk.

**Questions:**

See "Weakness" and "Limitations".

**Limitations:**

The limitations are discussed in the paper. However, I believe there could be additional limitations that worth consideration.

For example, the experiment is very limited, only CIFAR datasets are tested, which contains tiny images only. It would be desirable to understand how the gradient inversion attack work on moderate images (e.g. CelebA or ImageNette) and even maybe on language samples.

**Strengths And Weaknesses:**

Strength:

Overall the paper is well-written and tackles a clear task. The FL privacy is an important and practical concern, especially in the black-box setting in which the authors have worked on. The motivation of an honest-but-curious host could be realistic. Empirically speaking, the ablation study of BN modes, number of channels, skip connection, etc. is thorough. From a theoretical viewpoint, the proposed angular Lipschitz constant has nice properties (scale-invariance); the theorem has a premise reasonably justified.

Weakness:

My main concern is the impact of this paper: the experiment is limited to one simple dataset, thus limiting my confidence that the trends observed in this paper can hold in a broader range of cases (see more comments in "Limitations"). All conclusions except the scale-invariant property of the new Lipschitz measure are empirical, which requires more experiments to support the trends claimed in the paper.

---

> ### Author Response · Authors · 2022-08-02
> **Rebuttal for R2's comments**
>
> # W1 : More experimental validation of the proposed measure, angular Lipschitz smoothness
>
> Thanks for appreciating the contribution of our work first. We added experimental results on ImageNet (high-resolution image), other architectures like MobileNet v2 and ShuffleNet variations, and self-supervised models. Please see the appendix for the update. We experimentally showed our measure can thoroughly act as an upper bound of reconstruction quality at any setting, thus increasing the applicability of our measure.

---

> ### Author Response · Authors · 2022-08-08
> **R2 - The discussion period closes in about 24 hours**
>
> Today is the end of the discussion deadline. Could you please go over our rebuttal and check the responses? We believe that we have addressed all your concerns and that including these discussions will further strengthen our paper. We hope you reflect this in your final review and the score. We thank you again for your time and efforts in reviewing our paper.
>
> Thank you,
> Authors.

---

### Official Review · Reviewer_f87g · 2022-07-12

**Rating:** 3
**Confidence:** 4
**Soundness:** 1 poor
**Presentation:** 3 good
**Contribution:** 2 fair

**Summary:**

This paper first evaluate current gradient inversion attacks for both implicit and explicit model changes. It shows that when the batch norm mode, training epochs, skip connection or channel size is different, the result of gradient inversion attacks also changes. Finally, it proposes angular lipschitz smoothness and shows its positive correlation to the success of gradient inversion attacks.

**Questions:**

See weaknesses for details.

**Limitations:**

Yes, the authors fairly describe the limitations.

**Strengths And Weaknesses:**

### Strengths
1. It is the first paper to study gradient inversion attacks on the different settings of training procedures and models, including batch-norm mode, skip connection and channel size.
2. Its second part tries to find another signal, which is able to indicate the leakage in the end. Such signal, if it has good performance, would be useful in practice.
3. The paper is well structured.

### Weaknesses
The main weakness of this paper is that some experiment results are not very reliable and and some conclusions from the results are ambiguous.
1. Results of BN:
    (a). Why all MSE numbers in table 1 are mostly larger than 1? If all images are lying in the 0-1 cube, MSE > 1 means nothing recovered.  However, results from previous work already shows the success of image recovery at some setting of BN.
    (b). What is the conclusion from the BN experiment results?
2. Results of skip connection: why the gradient inversion attack is failed for the ConvNet? [1] and its following work all shows the success for ConvNet.
3. Results of channel size: the result "the reconstruction quality worsens as the number of channels increases for BN set to the train mode" is lack of explanation.
4. Angular Lipschitz smoothness: The intuition from theorem 1 is that smaller L means faster dropping at one step of optimization. However, it doesn't mean the minimum value of L_grad^x would be smaller, i.e. the reconstruction result is better. As the experiment details are missed in both main text and appendix, I am hypothesizing it is likely due to the unfair optimization set-up, e.g. the larger L_grad might need more iterations to converge, etc.

As for the experiment set-up, what gradient inversion attack is evaluated for those experiments?

[1] Zhu, Ligeng, Zhijian Liu, and Song Han. "Deep leakage from gradients." Advances in neural information processing systems 32 (2019).

-------After Discussion--------

I have read all responses and I'll keep my score. Thank the authors for the clarification and the additional experiment results. I hope authors can improve their paper by incorporating these results in the revision and improve their writing by highlighting the conclusions for each setting (BN, skip connection, training stages etc.) with the clear conditions.

---

> ### Author Response · Authors · 2022-08-02
> **Rebuttal for R1's comments**
>
> # W1 : Results of BN - why MSE > 1? Previous work already shows the success of image recovery at some setting of BN. The conclusion?
>
>
> MSE > 1 is due to image normalization (i.e., MSE is computed on normalized images). With the pre-computed channel-wise mean \mu and channel-wise standard deviation \sigma over training dataset, image x is normalized to (x - \mu)/\sigma. Therefore, the normalized image lies in the [-\mu/sigma, (1-\mu)/sigma] cube, not the 0-1 cube mentioned by the reviewer 1. Actually, for CIFAR100 datasets, \sigma = [0.267, 0.256, 0.276]. Therefore, MSE=1.8 in Table 1 of our original paper can be re-interpreted as approximately sqrt(1.8*(0.26)^2) = 0.349 pixel difference in 0-1 scale. For this case (MSE=1.8), some images are partially recovered thus not completely unrecoverable. The reviewer 1 seems to refer to [10] as a successful recovery case at some setting of BN. However, they use different experimental options compared to ours. For example, they used the dataset CIFAR10 and trained ResNet18 model on CIFAR10 for 200 epochs with learning rate decay 0.1 for every 50 epoch while we used CIFAR100 and trained ResNet 18 model on CIFAR100 for 300 epochs with learning rate decay 0.1 at epoch 150 and 225 (standard version of ResNet training). Our BN experiments suggest that reconstruction quality can differ completely depending on bn mode for gradient computation at an early stage of FL training (please see Figure 2.).
>  In addition, our code is modified from the original code by the previous work but the MSE computation part is unchanged.
> (The link for the GitHub page: https://github.com/JonasGeiping/invertinggradients).  This code has been widely used for follow-up works, thus our MSE computation is trustable.
>
> # W2 : Results of skip connection
>
>
> Please remember that our experimental analysis is based on ‘batch’ reconstruction not a single image reconstruction, which is mentioned thoroughly in the abstract section and the introduction section of our main paper. Previous works [6, 10, 12, 26] consider ConvNet (a kind of architecture without skip connection) only for single image recovery, not batch recovery. [6, 10, 12, 26] focus on only ResNet architectures for batch recovery experiments. To our knowledge, this work is the first time to apply gradient inversion attack to ConvNet for recovery of multiple images (batch size = 16). After tuning some hyperparameters such as learning rate, \alpha_{tv}, and \alpha_{BN}, we obtained reasonable results for ConvNet, but still worse quality than that from ResNet models. Please see Table 6. of the updated appendix.
>
>
> # W3 : Results of channel size
>
> Our re-evaluation results motivate us to find some explanation for each result and we provided a unified and theoretically-grounded explanation with our proposed measure, angular Lipschitz smoothness. Please compare (c) of Figure 7. (resnet variations with BN train mode) to (f) of Figure 7. (resnet variations with BN eval mode). When BN layers are set to eval mode, L becomes smaller with increasing channels on average while there is no such trend in the BN train mode case.
>
>
>
>
> # W4 : Angular Lipschitz smoothness
>
> Theorem 1 of the main paper states that one-step loss drop is related to Lipschitz smoothness of gradient. Because reconstruction is about reaching to ground-truth image, we intend to compute Lipschitz smoothness around ground-truth, thus we naturally link our measure to reconstruction quality. Our theorem is not about final loss but we hope to prove a theorem regarding final training loss in future work. R1 suggests longer training epochs for models with large Lipschitz smoothness and we conducted the same experiments but for ten times longer epochs. However, the loss drop for models with large Lipschitz smoothness was marginal. Instead, we searched for an optimal initial learning rate (e.g. among 0.1, 0.01, and 0.001) for each model and it was more efficient and effective than a longer epoch setting.
>
> # Q1 : The experimental setup
>
> The target attack of our experiments is [10] with the assumption of the attacker knowing correct label information and exact batch statistics (one of the settings considered in [7]). However, please note that our BN mode analysis is focused on gradient computation rather than statistics variation and this work is the first to consider gradient inversion attack of models during training. Furthermore, we explain our re-evaluation results with our proposed measure derived from our theory. Our proposed measure correlates with an upper bound of reconstruction quality. This trend is thoroughly managed on even high-resolution dataset (ImageNet), memory-friendly models (MobileNet v2 and ShuffleNet variations), and self-supervised models. Please check our updated results in the appendix (supplementary material).

---

> > ### Comment · Reviewer_f87g · 2022-08-04
> > **Response**
> >
> > Thank author's effort for the clarification. I still have some questions and comments:
> >
> > w1: [1] shows that the batch norm will weaken the attack performance, which shares the similar conclusion to here.
> >
> > w2: Thanks for clarifying the setting multi-batch. My concern then would be: is it possible that the difference of results comes from the different numbers of parameters instead of "skip connection" when comparing ConvNet and ResNet? The more reasonable experiment would be delete the skip connection in ResNet and compare this variant of ResNet with the original ResNet.
> >
> > W4: How does the results change after searching the best learning rate?
> >
> > Q1: As the literature of gradient inversion mostly vary in the optimization objectives, what is the optimization objective for this paper?
> >
> > Comment: It is better to describe the experimental set-up as clarified in the reply, as the conclusions and results depends on the set-up, e.g. the normalization of image space for MSE and the multi-batch setting for skip-connection experiments.
> >
> > [1] Huang, Yangsibo, et al. "Evaluating gradient inversion attacks and defenses in federated learning." Advances in Neural Information Processing Systems 34 (2021): 7232-7241.

---

> > > ### Author Response · Authors · 2022-08-05
> > > **Rebuttal for the first discussion from Reviewer f87g (R1)**
> > >
> > > ## w1 : The conclusions of [1] and our work
> > >
> > > Neither [1] (R1's reference) nor our paper concludes that batch norm will weaken the attack performance, which is not true in reality.
> > >
> > > In [1], most experiments consider the gradient inversion attack with BN statistics matching loss to consider stronger attack in the evaluation. The conclusion of [1]'s BN experiments is that using exact batch statistics provides better reconstruction quality than using running statistics (BN proxy) or inferred statistics (BN infer), which is about how exactness of statistics affects the reconstruction quality.
> > >
> > > In ours, most experiments are based on gradient inversion attack with BN statistics matching loss to consider stronger attack as in [1].However, we focus on how BN modes affect reconstruction quality, in terms of gradient computation, not BN statistics. In our experiments, we always use exact batch statistics (for stronger attack according to [1]) for both BN train mode and eval mode. Again, our BN experiments are not about BN statistics, which is the focus of [1], or the existence of BN statistics matching loss.
> > >
> > > ## w2 : resnets after skip connection removal instead of ConvNet
> > > Thanks for the valuable comment. We agree that deleting skip connections in resnets would be a fair ConvNet counterpart. We were unsure about whether such an architecture can be successfully trained or not. Note that we consider training iterations to see how the attack method applies during training of FL. We'll try it now and update the result soon. Please stay tuned.
> > >
> > > ## w4 : The results with optimal learning rate
> > >  Please see the updated results (Table 5-8) in the appendix of supplementary material.
> > >  Finding an optimal learning rate was effective in raising the reconstruction quality over  different L values. However, for large L, optimization for gradient matching was still difficult due to small loss drop and small optimal step size. Please see Figures 21-22 in the appendix. The plots describe how gradient matching loss changes by small learning rate (1e-5) optimization near ground truth (initial loss is close to zero) depending on L in the version of SGD optimizer (Figure 21) and Adam optimizer (Figure 22). In each plot, the estimated image easily escapes from nearby ground truth even with small learning rate for large-L models. This implies that models with large L rarely converge to ground truth.
> > >
> > >
> > > ## Q1 : Our optimization objective is like the following:
> > >
> > >       Gradient matching loss : cosine-similarity loss
> > >       BN statistics loss : Yes, we have
> > >       BN stats : exact BIN statistics (for stronger attack)
> > >       Batch size : 16 for CIFAR100, 10 for CIFAR10, and 1 for ImageNet (batch recovery     attack for high-resolution image is not reproduced yet)
> > >       Private labels : Known, we assume an attacker already knows correct label information for stronger attack.
> > >       Image normalization : Yes, image is normalized with channel-wise mean and standard deviation. MSE is computed in this normalized box.
> > >
> > > Our major experiments are based on the setting above. However, we partially observed that our measure can thoroughly act as an upper bound of reconstruction quality in the setting of L2 gradient matching loss or the setting of BN loss exclusion.
> > >
> > > **If our rebuttal addresses your concerns, we hope you reflect this in your final review and score. We thank you again for your time and efforts in reviewing our paper and having discussions that strengthen our paper.**
> > >
> > > [1] Huang, Yangsibo, et al. "Evaluating gradient inversion attacks and defenses in federated learning." Advances in Neural Information Processing Systems 34 (2021): 7232-7241.

---

> > > ### Author Response · Authors · 2022-08-07
> > > **The updated results for w2 : skip connection-removed resnets**
> > >
> > > We added the results for skip connection-removed ResNet18 mentioned by R1 in Table 9. of the updated appendix.
> > > Table 9. experimentally shows that LPIPS of all the reconstructed images from skip connection-removed ResNet18 models is greater than 0.5.
> > > As described in Figure 9., LPIPS > 0.5 implies that the detailed information of original images is rarely contained in the reconstructed image.
> > > The reconstruction quality was not good even for no-skip connection counterparts (fairer ones mentioned by R1) of ResNet18 in our experiments.

---

> > > > ### Comment · Reviewer_f87g · 2022-08-08
> > > > **Response**
> > > >
> > > > Thanks for your additional experiment results. By comparing Table 9 and Table 1, I am wondered why the MSE of reconstructed images without skip connection-removed ResNet18 is consistently better than the one with vanilla ResNet18, which has the different conclusion from LPIPS.

---

> > > > > ### Author Response · Authors · 2022-08-08
> > > > > **Rebuttal for the second discussion from Reviewer f87g (R1)**
> > > > >
> > > > > Please see Table 5 for the results of ResNet18 models (optimal learning rate version) instead of Table 1 (older version in the original submission).
> > > > >
> > > > > Then, reconstructed images from skip connection-removed ResNet 18 is sometimes better or sometimes worse than vanilla ResNet 18 in terms of MSE.
> > > > >
> > > > > In addition, there is no need for MSE to perfectly correlate with LPIPS since MSE is the distance in image space while LPIPS is the distance in embedding space of VGG networks.
> > > > >
> > > > > We just chose LPIPS as a measure for image similarity to follow the trend in previous works [1,2,3]. MSE is rather rarely studied as a measure for image similarity in previous works.
> > > > >
> > > > > Our results in terms of MSE show that our proposed measure, L-cos can correlate with an upper bound of reconstruction quality in MSE also (implied from the tables). We plan to put all the results for all the measures together in the future.
> > > > >
> > > > >
> > > > > [1] Huang, Y., Gupta, S., Song, Z., Li, K., & Arora, S. (2021). Evaluating gradient inversion attacks and defenses in federated learning. Advances in Neural Information Processing Systems, 34, 7232-7241.
> > > > >
> > > > > [2] Yin, H., Mallya, A., Vahdat, A., Alvarez, J. M., Kautz, J., & Molchanov, P. (2021). See through gradients: Image batch recovery via gradinversion. In Proceedings of the IEEE/CVF Conference on Computer Vision and Pattern Recognition (pp. 16337-16346).
> > > > >
> > > > > [3] Jeon, J., Lee, K., Oh, S., & Ok, J. (2021). Gradient inversion with generative image prior. Advances in Neural Information Processing Systems, 34, 29898-29908.

---

> ### Author Response · Authors · 2022-08-08
> **R1 - The discussion period closes in about 24 hours**
>
> First of all, thanks for R1's helpful comments and discussions.
>
> Thanks to R1's discussion on convergence of gradient matching loss, we could obtain better results with optimal learning rate during rebuttal.
> Based on improved results, we are certain that image reconstruction mostly fails for models with large L-cos (angular Lipschitz smoothness), which supports the role of our proposed measure as an upper bound of reconstruction quality (in terms of both MSE or LPIPS).
>
> We also checked that, when estimated images are initialized very close to ground-truth images, the reconstructed solution turns out to be easily far from ground truth for models with large L-cos even with very small learning rate. This additional analysis also supports our idea.
>
> In addition, please consider our extensive demonstration of our proposed measure on high-resolution images (ImageNet), other architectures (MobileNetV2, ShuffleNetV2 variations), and even self-supervised models , along with theoretical proof in the original submission.  **We believe that our rebuttal addresses all your concerns and hope you reflect this in your final review and the score.**

---

> ### Author Response · Authors · 2022-08-09
> **R1 - The discussion period closes in about 3 hours**
>
> Today is the end of the discussion deadline. Could you please check the additional responses? We believe that we have addressed all your concerns and that including these discussions will further strengthen our paper. **We hope you reflect this in your final review and the score. We thank you again for your time and efforts in reviewing our paper.**
>
> Thank you,
> Authors.

---

### Author Response · Authors · 2022-08-02
**Common comment 1**

We first designate the reviewer numbers as follows.

R1: Reviewer f87g
R2: Reviewer 9iUp
R3: Reviewer vsQD
R4: Reviewer F6kS


# C1. Our definition of black-box setting in federated learning (FL) and its motivation [R3, R4]

The term ‘black-box’ might confuse some reviewers because ‘black-box’ usually means inaccessibility to even model parameters [1]. However, the term ‘black-box’ in our paper refers to cover-up of model architecture, not model parameters, from clients, thus disallowing full control of clients over model. In this conceptual setting, the server and clients transmit model parameters and gradient updates as a sequence of numbers (clients never know the meaning of each number due to lack of architecture or module information). Then, clients are dedicated to put model parameters and along with their data (as a non-modularized sequence of numbers) to the protected (thus unmodifiable) model (think of protected class concept in c++ language) of the client program and gradient will be popped up as an output. Our proposed setting is meaningful in two aspects: closed-source fashion and model inversion attack.

First, enterprises tend to share their technology as a closed source. For example, OpenAI announces that GPT-3 and DALL-E2 are released for commercial profits without sharing model information [2,3]. In the future, such hyperscale models can be exploited for FL to provide a more personalized (user-friendly) service.  In the case of FL, model weights should be shared, thus we chose to hide model architecture of the server. Instead of blaming closed-source technology, we suggested our ‘black-box’ setting as an adaptive, but conceptual form of FL.

At first glance, this setting can be unfair to clients due to lack of model architecture information, but we say that gradient information is sufficient for detecting whether the shared model is robust under known optimization-based gradient inversion attacks, with our proposed measure L-cos. The process of finding minimal information to clients would be important in establishing a middle ground between enterprises (servers) and clients. Furthermore, our ‘black-box’ setting has a side benefit that clients cannot infer some private information leaked in the model weights itself through model inversion attack [4,5], which is white-box.

Even in the ‘white-box’ setting, our measure is still meaningful since it is an efficient method for estimating reconstruction results of known gradient inversion attacks without directly applying them. Note that computing L-cos through random sampling requires only gradient computation while optimization-based gradient inversion attack requires gradient for gradient matching. As R3 pointed out, robustness against  known attacks is not sufficient for privacy guarantee. However, defense strategies are usually evaluated with robustness against known attacks, which is practically meaningful [6,7]. To clarify, we are meant to check model vulnerabilities against known attacks, not to guarantee absolute privacy.

---

> ### Comment · Reviewer_vsQD · 2022-08-04
> **Some questions**
>
> Ok, I now understand a bit better what the original goal with this black-box setting was. Yet, does this make sense? I would think that in this setting, privacy is impossible for the client device to analyze. The black-box application can trivially return the gradient update in any permutation, or as a stream of non-consecutive packages with error correction, and in general as encrypted data. It appears to me that a user that only observes data inflow/outflow into a black-box application could never be able to infer whether their privacy was broken.
>
> Further, that "defense strategies are usually evaluated with robustness against known attacks," is a problem in adversarial ML literature and does not reflect best practices in security analysis.

---

> > ### Author Response · Authors · 2022-08-05
> > **Some questions about 'black box' setting - server side gradient manipulation**
> >
> > We agree that encrypted gradient or gradient with randomized positions can be possible and this is a valuable thought that encourages the embodiment
> > of our conceptual 'black-box' setting.
> >
> > In our 'black-box' setting, we assume that no encryption is applied to gradients or that encryption allows clients to compute our proposed measure, angular Lipshictz smoothness.
> >
> > Our work is primarily focused on theoretically and empirically proposing a measure that explains the model’s vulnerabilities under optimization-based gradient inversion attacks.
> >
> > Therefore, please consider our work as the first step to consider the basic setting (no encryption) of ‘black-box’ (in our definition). We think your comments would be helpful for the concrete implementation of reliable protocol between the servers and clients
> >
> > On the other hand, please note that, even for the case of 'white-box' setting, our measure still acts as an efficient predictor of gradient inversion attacks, which might be useful when testing a large number of models.

---

### Author Response · Authors · 2022-08-02
**Common comment 2-1**

# C2. The novelty of the first part (model variations) in our main paper [R3, R4]

## Implicit variations: BN modes and training epochs

First, we focus on BN modes for gradient computation, rather than statistics variation. BN layers are versatile in terms of gradient computation, depending on BN modes. When BN layers are set to eval mode, batch mean and variance is constant, thus they are just channel-wise linear layers and gradient computation is simple channel-wise scaling. However, when BN layers are set to train mode, batch mean and variance are computed with learnable variables, thus backpropagation is more complicated.

To check the gradient difference, we conducted a simple experiment computing the average cosine similarity between gradients of the same batch of CIFAR-100 computed with BN eval mode and train mode on ResNet-18 model checkpoints of different epochs e (e = 0, 5, 10, 25, 50, 150, 225, 300). The average cosine similarity was as low as 0.1~0.3, which experimentally suggests the difference between gradient nature of models with BN eval mode and train mode. Therefore, we naturally intend to observe how the quality of reconstructed images from gradient inversion attacks differs depending on BN modes.

 For example, we found that known gradient inversion attacks rarely reconstruct images for untrained ResNet18 models with BN train mode, but succeed in recovering images somehow when turning BN layers to eval mode (see Figure 2. of our main paper). This trend is maintained even when BN statistics matching loss term is excluded, thus the difference of gradient nature between BN modes might be the reason for the phenomenon.

When it comes to BN statistics, we assume that clients send exact batch statistics for both BN eval mode and train mode to consider stronger attacks as discussed in [7]. Our BN eval mode does not mean clients sharing BN statistics as running statistics with the server. In our BN eval mode case, exact batch statistics are computed with BN eval mode forward .

In contrast to our work, previous works [7,8,9] focus on BN modes for statistics matching, rather than gradient computation. [9] is the first work to introduce BN statistics matching for strengthening gradient inversion attacks with the assumption of clients sharing exact batch statistics. The follow-up work [7] claims that the power of the attack considered in [9] can be mitigated when batch statistics is replaced with running statistics. Then, [8] suggests an adaptive gradient inversion attack as an attacker’s solution to the case of sharing running statistics as mentioned by R3.

Moreover,  [10, 7, 11] seem to use BN eval mode and [8.9] seem to use BN train mode for their gradient computation according to public github repositories [12,13,14] and our reproduction experiences. Again, we explore the difference between BN modes for gradient computation.

---

> ### Comment · Reviewer_vsQD · 2022-08-04
> **Relationship to Huang et al , "Evaluating Gradient Inversion Attacks and Defenses in Federated Learning"**
>
> In direct comparison to Huang et al., Sec.3.3, it is so far still unclear me what additional insights are generated in the batch norm discussion of this submission, aside from evaluating the claim over multiple training checkpoints.

---

> > ### Author Response · Authors · 2022-08-05
> > **Clarification on how our work relates to Huang et al.**
> >
> >  Huang et al. [1] only considers fully trained ResNet18-models for their evaluation but along with different levels of the exactness of batch statistics. (BN_exact, BN_infer, BN_proxy).
> > In contrast, the exactness of batch statistics is not dealt with in our paper and we only use exact batch statistics (BN_exact) for both BN eval mode and train mode to consider stronger attacks.
> >
> > In the case of fully trained ResNet18 models, reconstruction quality is not that different between BN eval mode and BN train mode. However, after considering different training epochs, we found that the models (ResNet variations, ConvNet) with BN train mode fail to reconstruct images, which are not the case in [1]. Therefore, this interesting phenomenon
> > motivates us to find a measure, Angular Lipschitz smoothness, that can explain the result of gradient inversion attacks. Please reconsider our theoretical validation and thorough experimental validation.
> >
> > [1] Huang, Y., Gupta, S., Song, Z., Li, K., & Arora, S. (2021). Evaluating gradient inversion attacks and defenses in federated learning. Advances in Neural Information Processing Systems, 34, 7232-7241.

---

### Author Response · Authors · 2022-08-02
**Common comment 2-2**

Second, to our knowledge, this work is the first to consider model checkpoints with different training epochs for gradient inversion attack. R3 mentioned that exploring model inversion attacks through different training epochs (or training states) is already discussed in previous works [10, 15, 16], which is not true.

To clarify our novelty, we want to explain the relation of our work to previous works mentioned by R3 and R4. First, [10] considers only fully trained(converged to training set) or untrained models for attack. [15] considers models with different training states, but fully trained with different degrees of perturbation to gradients (as a defense strategy).

However, we are curious about previous works’ setting that model inversion attacks are tested at the end of training of a large training set, or completely before training [10, 15, 16]. Why not gradient inversion attacks happen during training? This question motivates us to re-evaluate gradient inversion attacks on models with different training epochs and we found that reconstruction can fail for untrained ResNet models when BN layers are set to BN train mode and the best recovery quality can be obtained at the very early stage of training (see Figure 2. of our main paper).

 For [16], the term ‘local training iterations’ means the number of local model updates at the clients’ device before sending gradients to the server, not training epochs of global model. [16] also considers fully trained models for the evaluation.

 Furthermore, our experimental result breaks the previous belief from [10] that gradient is small for trained models and models with small gradient (or small information) might be robust under gradient inversion attacks (please see the ‘label flipping attack’ part of [10]). We found that gradient can be larger for trained models compared to untrained counterpart and reconstruction even fails for such larger gradient of trained models (See (a), (d) of Figure 7.).  This counterintuitive phenomenon can be explained by our theorem (Theorem 1) and experimental results as the language of (angular) Lipschitz smoothness (See Figure 7.).

---

> ### Comment · Reviewer_vsQD · 2022-08-04
> **Clarification**
>
> I'm sorry to have provided an mistaken reference here, concerning evaluations for intermediate training checkpoints (without batch normalization), the reference should be Mo et al, "Quantifying Information Leakage from Gradients", https://arxiv.org/abs/2105.13929v1, see Fig. 1 and Fig.7.

---

> > ### Author Response · Authors · 2022-08-05
> > **The clarified reference [1] about training epochs**
> >
> > Yes, we agree that [1] conducts an analysis on how training epochs affect information leakage but their results are partial compared to our experimental findings.
> >
> > In [1], information leakage decreases as training progresses like the case of models with BN eval mode in our results (see Tables 5-8 of the appendix in the updated supplementary material).
> >
> > On the other hand, we found that untrained models with BN train mode fail to reconstruct images from their gradients, which is the firstly discovered one in our work.
> >
> > We plan to cite [1] for the acknowledgement of their first consideration of training epochs in the evaluation of information leakage from gradients, but please note that our novel finding is not included in [1].
> >
> >
> > [1] Mo, F., Borovykh, A., Malekzadeh, M., Haddadi, H., & Demetriou, S. (2021). Quantifying information leakage from gradients. CoRR, abs/2105.13929.

---

### Author Response · Authors · 2022-08-02
**Common comment 2-3**

## Explicit variations : Skip-connection and channel size

As mentioned by R3, [10] is the first work to check the vulnerability under model inversion attack on convnet (the network without skip connection) and residual networks with different widths, but for a single image recovery, not a batch (multiple images). However, we focus on batch recovery, thus our thorough re-evaluation of gradient inversion attack on convnet and resnet variations is novel. For example, the previous work [10] discovers that resnet models with increased channel size are more vulnerable to gradient inversion attacks due to increased information. However, we found that this law does not apply to resnet models when BN layers are set to train mode. We explain this phenomenon through our proposed measure, (angular) Lipschitz smoothness. Please compare (c) of Figure 7. (resnet variations with BN train mode) to (f) of Figure 7. (resnet variations with BN eval mode). When BN layers are set to eval mode, L becomes smaller with increasing channels on average while there is no such trend in the BN train mode case.

---

### Author Response · Authors · 2022-08-02
**Common comment 3**

# C3. More experiments [R2, R3, R4]

As R2, R3 and R4 want more experimental validation of our proposed measure through other architectures and high-resolution datasets, we add our experimental results about the relationship between our measure and reconstruction quality on CIFAR10, CIFAR100, and ImageNet. For added experimental results, we tuned hyperparameters such as learning rate, \alpha_{tv}, and \alpha_{BN} for better attack quality.

Specifically, we focus on single image recovery for ImageNet, due to the lack of general batch recovery method for high-resolution images as discussed in the forum [17]. Note that [9] considers only few ResNet variations (three models) for ImageNet batch recovery and there is no github code for this attack. Instead, we experimented with other architectures, MobileNetv2 and ShuffleNetv2 variations, which are memory friendly for ImageNet.

 In addition, we consider several self-supervised ResNet50 models for ImageNet as suggested by R3 (different initialization schemes). We show that our measure can thoroughly act as an upper bound of reconstruction quality (i.e., reconstruction will fail when our measure exceeds some degree). We also conduct other experimental analyses to address some concerns from reviewers. All the additional experiments are included in the appendix (supplementary material). Please see the description below for a quick view of each experiment.

Please see the appendix of the updated supplementary material.

Tables 5-8 : The reconstruction quality results of ResNet18, ConvNet, ResNet18-2, and ResNet18-4 models on 64 CIFAR100 Images. Hyperparameters such as learning rate, $\alpha_{tv}$, and $\alpha_{BN}$ are tuned during rebuttal for better quality.

Table 9 : he reconstruction quality results of skip connection-removed ResNet18 models on 64 CIFAR100 Images. Hyperparameters such as learning rate, $\alpha_{tv}$, and $\alpha_{BN}$ are tuned during rebuttal for better quality.

Figure 9 : LPIPS of reconstructed images on CIFAR100. This result is for the readers to know how large LPIPS should be to remove detailed image information.

Figure 10 : LPIPS of reconstructed images on ImageNet. This result is for the readers to know how large LPIPS should be to remove detailed image information.

Figures 11-12 : The correlation plot for loss drop and reconstruction quality on 64 CIFAR100 images

Figures 13-14 : The correlation test for loss drop and reconstruction quality on 40 CIFAR10 images

Figures 15-16 : The correlation test for loss drop and reconstruction quality on 40 CIFAR10 images, but BN stats matching loss is excluded.

Figures 17-18 : The correlation test for loss drop and reconstruction quality on 3 ImageNet images. WRN50-2 [18], MobileNet_V2 [19], VGG19 [20], and ShuffleNet_V2 [21] models are added for evaluation.

Figures 19-20 : The correlation test for loss drop and reconstruction quality on 3 ImageNet images. Self-supervised models [22] are considered for the evaluation.

Figures 21-22 : The correlation test for loss drop for initialized images near to ground-truth. SGD and Adam optimizers with initial learning rate 1e-3 are considered for each figure. Models with higher L tend to escape from ground-truth.

---

### Author Response · Authors · 2022-08-02
**Local indices of references (different from those from main paper)**

Here are local indices of reference papers for the rebuttal.

[1] Peter Kairouz, Brendan H. Mcmahan, Brendan Avent, Aurélien Bellet, Mehdi Bennis, et al.. Advances and Open Problems in Federated Learning. Foundations and Trends in Machine Learning, Now Publishers, 2021, 14 (1-2), pp.1-210. ffhal-02406503v2f

[2] The reddit forum about closed-source GPT-3 (Link : https://www.reddit.com/r/LanguageTechnology/comments/iz0iyw/what_is_your_opinion_regarding_gpt3_being_closed/).

[3] The article about commercial use of DALL-E 2 (Link : https://techcrunch.com/2022/07/20/openai-expands-access-to-dall-e-2-its-powerful-image-generating-ai-system/?guccounter=1&guce_referrer=aHR0cHM6Ly93d3cuZ29vZ2xlLmNvbS8&guce_referrer_sig=AQAAAKZxXCEUI9Wf0kvN5UT6BGEOaHFzD-E-jdGSKLP0OfETd5RoLRoLSLt0Ooe8pUIEdhTWmYXV5l0lkPjhuXAu3QunUxv7kdoOXAbKziom-8aGyE0hHmRz26mSIyRdwVERfHP7EOLzs8gSHPjXnBi7YgmSqT2U4f-aV-gU4MDgMGIw).

[4] Wang, K. C., Fu, Y., Li, K., Khisti, A., Zemel, R., & Makhzani, A. (2021). Variational Model Inversion Attacks. Advances in Neural Information Processing Systems, 34, 9706-9719.

[5] Zhang, Y., Jia, R., Pei, H., Wang, W., Li, B., & Song, D. (2020). The secret revealer: Generative model-inversion attacks against deep neural networks. In Proceedings of the IEEE/CVF conference on computer vision and pattern recognition (pp. 253-261).

[6] Huang, Y., Song, Z., Li, K., & Arora, S. (2020, November). Instahide: Instance-hiding schemes for private distributed learning. In International conference on machine learning (pp. 4507-4518). PMLR.

[7]  Huang, Y., Gupta, S., Song, Z., Li, K., & Arora, S. (2021). Evaluating gradient inversion attacks and defenses in federated learning. Advances in Neural Information Processing Systems, 34, 7232-7241.

[8] Hatamizadeh, A., Yin, H., Molchanov, P., Myronenko, A., Li, W., Dogra, P., ... & Roth, H. R. (2022). Do Gradient Inversion Attacks Make Federated Learning Unsafe?. arXiv preprint arXiv:2202.06924.

[9] Yin, H., Mallya, A., Vahdat, A., Alvarez, J. M., Kautz, J., & Molchanov, P. (2021). See through gradients: Image batch recovery via gradinversion. In Proceedings of the IEEE/CVF Conference on Computer Vision and Pattern Recognition (pp. 16337-16346).

[10] J. Geiping, H. Bauermeister, H. Dröge, and M. Moeller. Inverting gradients-how easy is it to break privacy 331 in federated learning? Advances in Neural Information Processing Systems, 33:16937–16947, 2020.

[11] J. Jeon, K. Lee, S. Oh, J. Ok, et al. Gradient inversion with generative image prior. Advances in Neural 344 Information Processing Systems, 34:29898–29908, 2021.

[12] The github repository for the work [10] (Link for the evidence that they used only BN eval mode : https://github.com/JonasGeiping/invertinggradients/issues/6).

[13] The github repository for the work [7] (Link for the default setting as BN eval mode:
https://github.com/Princeton-SysML/GradAttack/blob/master/gradattack/attacks/gradientinversion.py).

[14] The github repository for the work [11] (Link for the default setting as BN eval mode: https://github.com/ml-postech/gradient-inversion-generative-image-prior/blob/main/rec_mult.py).

[15] L. Zhu, Z. Liu, and S. Han. Deep leakage from gradients. Advances in Neural Information Processing Systems, 32, 2019.

[16] Wei, W., Liu, L., Loper, M., Chow, K. H., Gursoy, M. E., Truex, S., & Wu, Y. (2020). A framework for evaluating gradient leakage attacks in federated learning. arXiv preprint arXiv:2004.10397.

[17] The openreview forum of the paper [7] (Link : https://openreview.net/forum?id=0CDKgyYaxC8).

[18] Zagoruyko, S., & Komodakis, N. (2016). Wide residual networks. arXiv preprint arXiv:1605.07146.

[19] Sandler, M., Howard, A., Zhu, M., Zhmoginov, A., & Chen, L. C. (2018). Mobilenetv2: Inverted residuals and linear bottlenecks. In Proceedings of the IEEE conference on computer vision and pattern recognition (pp. 4510-4520).

[20] Simonyan, K., & Zisserman, A. (2014). Very deep convolutional networks for large-scale image recognition. arXiv preprint arXiv:1409.1556.

[21] Ma, N., Zhang, X., Zheng, H. T., & Sun, J. (2018). Shufflenet v2: Practical guidelines for efficient cnn architecture design. In Proceedings of the European conference on computer vision (ECCV) (pp. 116-131).

[22] Ericsson, L., Gouk, H., & Hospedales, T. M. (2021). How well do self-supervised models transfer?. In Proceedings of the IEEE/CVF Conference on Computer Vision and Pattern Recognition (pp. 5414-5423).

---

### Meta-Review · Area_Chair_ZWuB · 2022-08-22

**Recommendation:** Reject
**Confidence:** Certain

**Metareview:**

Several analyses provided in the paper are not novel and known in the literature. The black-box setting is not properly motivated and impractical. The angular Lipschitz constant is a good contribution, but it seems to be only a small part of the paper. For these reasons, the reviewers are not convinced that the contribution of this paper is significant enough.

**Award:**

No

---

### Decision · Program_Chairs · 2022-09-14

Reject